# Epidemiologic Characteristics of Children with Diabetic Ketoacidosis Treated in a Pediatric Intensive Care Unit in a 10-Year-Period: Single Centre Experience in Croatia

**DOI:** 10.3390/medicina58050638

**Published:** 2022-05-05

**Authors:** Kristina Lah Tomulić, Lucija Matko, Arijan Verbić, Ana Milardović, Srećko Severinski, Ivana Kolić, Kristina Baraba Dekanić, Senada Šerifi, Ivona Butorac Ahel

**Affiliations:** 1Pediatric Intensive Care Unit, Clinical Hospital Center Rijeka, 51000 Rijeka, Croatia; arijan.verbic@gmail.com (A.V.); milarda9@yahoo.com (A.M.); 2Faculty of Medicine, University of Rijeka, 51000 Rijeka, Croatia; lucija.matko82@gmail.com (L.M.); sreckoseverinski@yahoo.com (S.S.); k.baraba.dekanic@gmail.com (K.B.D.); sesehr@gmail.com (S.Š.); ivonabuah@gmail.com (I.B.A.); 3Department of Pediatrics, Clinical Hospital Center Rijeka, 51000 Rijeka, Croatia; ikolic3107@gmail.com

**Keywords:** children, diabetic ketoacidosis, epidemiology, pediatric intensive care

## Abstract

*Background and Objectives*: The incidence of severe and moderate forms of DKA as the initial presentation of type 1 diabetes mellitus (T1D) is increasing, especially during the COVID-19 pandemic. This poses a higher risk of developing cerebral edema as a complication of diabetic ketoacidosis (DKA), as well as morbidity and mortality rates. The aim of this study was to determine the trend and clinical features of children treated in the last 10 years in the Pediatric Intensive Care Unit (PICU) due to the development of DKA. *Materials and Methods*: This retrospective study was performed in the PICU, Clinical Hospital Centre Rijeka, in Croatia. All children diagnosed with DKA from 2011–2020 were included in this study. Data were received from hospital medical documentation and patient paper history. The number of new cases and severity of DKA were identified and classified using recent International Society for Pediatric and Adolescent Diabetes (ISPAD) guidelines. *Results*: In this investigation period, 194 children with newly diagnosed T1D were admitted to our hospital: 58 of them were treated in the PICU due to DKA; 48 had newly diagnosed T1D (48/58); and ten previously diagnosed T1D (10/58). DKA as the initial presentation of T1D was diagnosed in 24.7% (48/194). Moderate or severe dehydration was present in 76% of the children at hospital admission. Polyuria, polydipsia, and Kussmaul breathing were the most common signs. Three patients (5.2%) developed cerebral edema, of whom one died. *Conclusions*: During the investigation period a rising trend in T1D was noted, especially in 2020. About one quarter of children with T1D presented with DKA at initial diagnosis in western Croatia, most of them with a severe form. Good education of the general population, along with the patients and families of children with diabetes, is crucial to prevent the development of DKA and thus reduce severe complications.

## 1. Introduction

Diabetic ketoacidosis (DKA) is a common and serious complication of diabetes mellitus in children. It is caused by a deficiency of circulating insulin and increased levels of the counterregulatory hormones, and is characterized by hyperglycemia, acidosis, and ketonemia/ketonuria. This medical emergency primarily affects children with type 1 diabetes mellitus (T1D); however, pediatric patients with type 2 diabetes may also develop DKA [1,2].

DKA is highly prevalent in both newly diagnosed and children previously known to have diabetes mellitus. The incidence of DKA at initial diagnosis ranges from 15% to 70% [1,3,4,5]. If T1D is detected early and prompt treatment is started, the progression to DKA will be avoided. Children previously known to have T1D develop DKA in 1–10% of cases [1,6,7].

DKA is associated with various clinical signs and symptoms, including dehydration, tachypnea, Kussmaul’s breathing, tachycardia, acetone breath, nausea, vomiting, abdominal pain, and decreased levels of consciousness of varying degrees. The most common cause of acute deterioration in children with DKA, and the leading cause of death, is cerebral edema. Clinically significant cerebral edema develops in <1% and is associated with the severity of the DKA, higher blood urea nitrogen and greater acidosis [8].

Recently published studies show large variations between countries in the incidence of DKA as the initial presentation of T1D [2,5]. There is also an increase in the incidence of T1D in children, as well as a growing trend of severe forms of DKA, especially during the COVID-19 pandemic [9,10,11].

When treating children with DKA in our PICU, we strictly follow the latest International Society for Pediatric and Adolescent Diabetes (ISPAD) guidelines [1]. The aim of this study was to determine the incidence, clinical features and trend during initial treatment in children admitted during the last 10 years in the PICU due to the development of DKA, and to compare our results with other centers in Croatia and other countries, as we have not had these results previously. We wanted also to investigate whether there was an increase in the number and severe forms of DKA in our center during the pandemic period.

## 2. Materials and Methods

The subjects of this retrospective study were children aged 0 to 18 years. All were hospitalized in the PICU, Clinic of Pediatrics, University Hospital Center Rijeka, between 1 January 2011 and 31 December 2020.

This PICU is one of four Level III PICUs in Croatia and cares for all critically ill children from three counties: Primorsko-goranska, Istarska, and Ličko-senjska. It covers the western part of Croatia, which is about 20% of Croatian territory. All children from these three counties with newly diagnosed T1D and with mild, moderate, and severe DKA are treated in our hospital. The clinical signs and laboratory analyses were recorded by pediatric intensivists.

The diagnosis of DKA was based on the recent ISPAD guidelines: hyperglycemia (blood glucose value ≥ 11 mmol/L), metabolic acidosis (venous blood pH < 7.3 or serum bicarbonate values < 15 mmol/L), and ketosis (presence of ketones in the blood or urine) [1]. The indications for admitting a child to the PICU were, in addition to laboratory findings, a disturbed general condition and particularly changes in consciousness level.

The level of consciousness was defined by the AVPU scale, where A indicates that the patient is awake and conscious (alert), V that they respond to a call (voice), P that they respond to a painful stimulus (pain), and U indicates that they do not respond to any stimuli (unresponsive).

Data for the study were collected retrospectively by pediatric intensivists from the Integrated Hospital Information System and the medical history of the above subjects. The total number of children with newly diagnosed T1D was collected by a pediatric endocrinologist. For each respondent with diagnosed DKA in the observed period, the following demographic and medical data were recorded: age, gender, family history, physical findings including level of consciousness, occurrence and duration of symptoms, and complications. The following laboratory findings were collected: blood glucose, blood pH, bicarbonate level (HCO_3_^−^), blood pCO_2_ level, base excess (BE), blood leukocyte and hemoglobin level, hemoglobin A1c (HbA1c) level, blood urea nitrogen (BUN), creatinine, sodium, and potassium. Blood glucose, pH, bicarbonates, and pCO_2_ values were measured every two hours during the stay in the PICU.

The study was approved by the Ethical committee of Clinical Hospital Centre Rijeka (approval number: 2170-29-02/1-20-2).

Statistical Analysis

The data were entered into electronic data tables via the computer program Microsoft Office Excel 365 and analyzed using the statistical software JASP 0.14.1.0. Nominal and ordinal measurements are shown through frequencies (*n*) and proportions (%), and numerically through average values and standard deviations. The correctness of the distribution was tested by the Shapiro–Wilk test. The structure of the observed variables is presented graphically and tabularly.

## 3. Results

T1D was newly diagnosed in 194 children in the observed period. All were admitted to the Department of Pediatrics, Clinical Hospital Centre Rijeka in Croatia. DKA as the initial presentation was present in 24.7% (Figure 1).

Due to DKA, 58 children were treated in PICU; 29 (50%) of these were girls. Most of them had newly diagnosed T1D (48/58, 82.8%), while the rest had previously diagnosed T1D (10/58, 17.2%).

During the 10-year investigation period, the greatest number of children with T1D were admitted in 2020 (*n* = 30); also, the greatest number of children with DKA were admitted to the PICU in the same year (12/58, 20.6% of all cases). The Mann–Kendall trend test does not show a change in DKA over the period 2011–2019 (tau = −0.0962, *p* = 0.6301). The same result is obtained for T1D (tau = 0.1110, *p* = 0.7545).

The average number of children with T1D in the period 2011–2019 was 18.2222 ± 4.2947, while in 2020 the number of cases was 30. T-test shows that this value deviates statistically significantly from the average (*p* < 0.0001).

The average number of children with DKA in the period 2011–2019 was 4.1111 ± 2.1473, while in 2020 the number was 11. T-test shows that this value deviates statistically significantly from the average (*p* < 0.0001) (Figure 1).

Most of the children with DKA admitted to the PICU were over nine years old (*n* = 28; 48.3%), and all children with DKA due to previously known T1D are in this age group (10/28). Six children under the age of two with T1D were admitted in the observed period; three of them (50%) had DKA (Figure 2).

Demographic data, symptoms and laboratory data at admission are presented in Table 1.

Close family members had T1D in 5.8% of cases (3/58), and type 2 diabetes (T2D) in 48.1% of cases (25/58). Polydipsia (74.1%), polyuria (70.7%) and Kussmaul’s breathing (56.9%) were the most common clinical signs in children admitted to the PICU with DKA. Other frequent clinical signs were weight loss (50%), fatigue (50%) and vomiting (43.1%). Laboratory findings for the children admitted to the PICU, analyzed in the first hour after admission, showed high glucose levels and low pH, bicarbonate concentration and serum carbon dioxide partial pressure (pCO_2_) level. There were also higher values of leucocytes and HbA1c. Defining the level of consciousness according to the AVPU scale, 77.6% (45/58) were awake and conscious at the time of admission, 15.5% (9/58) responded to voice, and 5.2% (3/58) responded only to pain stimuli. One child did not respond to any stimuli (1.7%).

At the time of admission to the PICU, 23 patients (40%) had clinical and laboratory signs of severe DKA, according to ISPAD guidelines from 2018. A moderate form of DKA affected 21 children (36%) and 14 were admitted due to a mild form (24%) (Figure 3).

During the first 24 h of treatment in the PICU, there were rapid changes in serum glucose concentration, pH, bicarbonate concentration, and serum carbon dioxide partial pressure levels. The fastest changes were recorded in the first eight hours of treatment (Figure 4).

In the investigation period, three children (5.2%) developed clinical signs of brain edema. Two of them recovered completely; one child died (1.7%).

## 4. Discussion

The percentage of DKA as the initial presentation of T1D varies significantly between institutions and countries. Recently published studies from centers in China, Australia, Saudi Arabia, and Serbia show large variations in the incidence of DKA as the initial presentation of T1D, with 50.1%, 48.1%, 37.7%, and 35.1%, respectively [5,12,13,14]. The lowest rates are reported in Sweden (14%), Denmark (14.7%), Canada (18.6%), and Finland (22%) [2,15,16]. In a study conducted in Croatia from 1995–2003, the incidence of DKA as the initial presentation of T1D for our region was 31.3% [17]. This wide variation in the frequency of DKA at onset of diabetes inversely correlates with the regional incidence of T1D and education of the population in recognition of early symptoms of the disease in children [1]. In our institution during the investigation period (2011–2020) the number of children with T1D increased slightly over the years (14–16/annually in 2011–2012 to 18–19/annually in 2018–2019 and to the top value of 30 in 2020). Meanwhile, except in 2020, there was no change in the incidence of DKA during these years and which remains around 25% in our hospital. Our results show a moderate incidence of DKA as the initial presentation of T1D, compared to other studies. It is very likely that better parental education about diabetes would contribute to a lower percentage of children admitted to the PICU due to this serious complication of T1D.

Stress is known to be one of the possible risk factors for developing T1D [18]. Coronavirus disease (COVID-19) during 2020 caused social distancing of children and adolescents and a sense of isolation and loneliness [18]. In addition, the viral disease itself can affect immune regulation and directly damage pancreatic beta cells [19]. A study conducted in Germany revealed an increase in T1D in children and adolescents at the time of the pandemic, but not a significant one compared to the period before the pandemic [20]. On the other hand, numerous papers have demonstrated a significant increase in DKA in its severe form as initial presentations of T1D during lockdown [21,22]. The cause of this is unknown, but it can be hypothesized that some of the causes may be public health measures, reduced face-to-face contact with primary care professionals and failure to recognize early symptoms of the disease during the pandemic period. Our results show a significant increase in new T1D cases in children and adolescents in 2020. Additionally, during 2020, 36.6% of children with newly diagnosed T1D were admitted to the PICU due to DKA, which is a significant rise compared to previous years. These data coincide with the results of the recent worldwide studies.

DKA can also develop in children with previously diagnosed T1D; the incidence is 1–10% [1]. The risk is higher in children who omit insulin, who had diarrhea and vomiting during viral infections, and in puberty mostly with alcohol consumption [23]. A recent meta-analysis shows a significant increase in the incidence of reported DKA in patients with continuous subcutaneous insulin infusion, mainly due to malfunction of insulin pump or catheter infusion [24]. In our study, 17.2% of children admitted to the PICU due to DKA had previously diagnosed T1D (10/58). All were older than 9 years, i.e., in prepubertal, pubertal, and/or postpubertal age. Comparing with the results of a study conducted by Burcul and co-workers in two other centers in Croatia, our results show a significantly lower percentage of DKA in children with previously known T1D [25]. For this group of patients, continuous education of children and family members in regular glycemic monitoring and recognition of early signs of ketoacidosis was crucial to prevent serious complications. For the last 10 years our institution provided a 24 h telephone consultation service with qualified nurses, which resulted in a relatively low number of DKA cases in children with previously known T1D.

Polydipsia and polyuria are early clinical signs of T1D, and in large clinical studies have been reported to occur in children in about 90% of cases [5,26]. These symptoms are usually present in children for about 12 to 25 days before diagnosis [26,27,28]. On the other hand, Kussmaul respiration is a late clinical sign of severe metabolic acidosis and a sign of advanced disease. Respiratory response usually develops within 24 h of the onset of metabolic acidosis. Respiratory compensation begins with tachypnea, but the progression of acidosis leads to the development of deep and rapid breathing and significant decrease in pCO_2_ [29].

In our study, the classical symptoms of polydipsia and polyuria were the most common clinical signs of T1D (Table 1). The next most common clinical sign, which occurred in almost 60% of the children with DKA, was Kussmail’s breathing. This coincides with the fact that in our study most children with DKA had severe metabolic acidosis on admission. Similar results were presented in the studies conducted in two other centers in Croatia [25]. Compared to the results from centers in China and Turkey (33.9% and 15.9%, respectively) our results show a higher number of children with severe metabolic acidosis on admission [5,26]. We can speculate that in our region if T1D symptoms are not recognized early by caregivers, when only polyuria and polydipsia are present, they are recognized very late when severe metabolic acidosis has developed and the risk of severe DKA complications is much higher.

According to the ISPAD guidelines from 2018 (and corresponding previous versions), emergency treatment of DKA should be started with an initial bolus of 0.9% saline 10 mL/kg over 30–60 min, if there are no signs of shock [1]. Additionally, if the blood glucose drops faster than 5 mmol/L/h after the initial bolus, adding glucose solutions should be considered due to the risk of sudden changes in blood osmolality and the possibility of developing brain edema. In our study, we monitored changes in glucose concentrations in children in the first 24 h, adhering strictly to the ISPAD guidelines. After the initial boluses of 0.9% saline in the first 60 min of treatment, in most cases we noticed a significant drop in blood glucose. This was almost always greater than 5 mmol/L/h. Therefore, we introduced glucose solutions very soon after the introduction of the insulin infusion, to slow down the glucose drop. It is considered that the time required to restore dehydration and to correct acidosis is about 11.6 ± 6.2 h [30]. In our study, the average time to reach pH > 7.3 was about 14 h, which was also the time when children were discharged from the PICU and transferred to the endocrinology department.

In pediatric intensive care and according to the Pediatric Index of Mortality (PIM 3), DKA is considered a low-risk diagnosis [31]. Although there is a need for intensive supervision and monitoring during the first 12 h in the PICU, the stay is usually short and children generally leave the intensive care unit in good condition. Nevertheless, the development of fatal brain edema is always possible despite strict adherence to the recommended guidelines. Although the development of brain edema in children with DKA, according to older published works, is less than 1% of cases, abnormalities in mental status occur in 4–15% [32,33]. Recent studies show that the incidence of brain edema in children with DKA might be much higher, 1.4%, 2.4%, 5.3%, and 5.9%, respectively [25,34,35,36]. However, further studies are needed to confirm this statement.

Despite adhering to the guidelines, three children (5.2%) at our facility developed signs of brain edema over a ten-year period and required urgent treatment with mannitol along with fluid reduction. All had a high risk for developing brain edema at admission, including severe DKA, extreme low pCO_2_, and bicarbonate along with high BUN. Two children fully recovered and had no neurological sequelae, while one child died of progressive brain edema.

The limitation of this study is the relatively small number of patients; with a larger sample the results might be somewhat different. In addition, it was a retrospective study in which data were collected from medical records and where the clinical condition of the child at admission depended on the personal assessment of the attending physician.

However, our research shows a rising trend in severe forms of DKA, and these worrying results should be confirmed in future studies. To determine the actual incidence of brain edema in children with DKA, research is also needed on a larger number of subjects.

## 5. Conclusions

There was a rising trend in cases of T1D in children, and consequently children with DKA, in the PICU in our institution, particularly in the pandemic year. Our study shows that in 24.7% of children DKA was the initial presentation of T1D. Most children in western Croatia present with a severe form of DKA, which requires treatment and careful monitoring in the PICU to identify and treat brain edema as early as possible and reduce morbidity and mortality. Since DKA and related complications in newly diagnosed T1D patients are entirely preventable with early recognition of the symptoms of diabetes, it is essential to carry out continuous campaigns and raise awareness of diabetes symptoms.

## Figures and Tables

**Figure 1 medicina-58-00638-f001:**
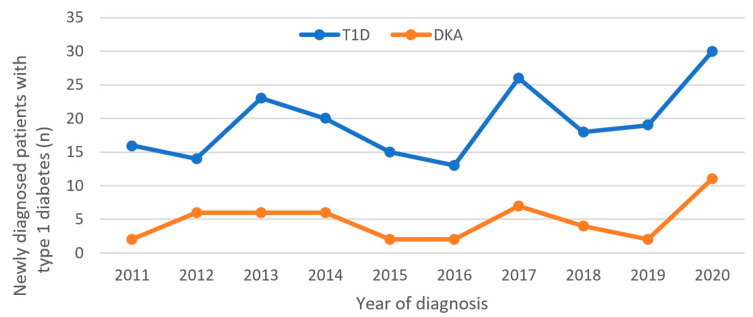
Number of children with newly diagnosed type 1 diabetes in Department of Pediatrics (Clinical Hospital Centre Rijeka, Croatia) in 2011–2020, and number of children with diabetic ketoacidosis at admission. Legend: T1D—type 1 diabetes; DKA—diabetic ketoacidosis.

**Figure 2 medicina-58-00638-f002:**
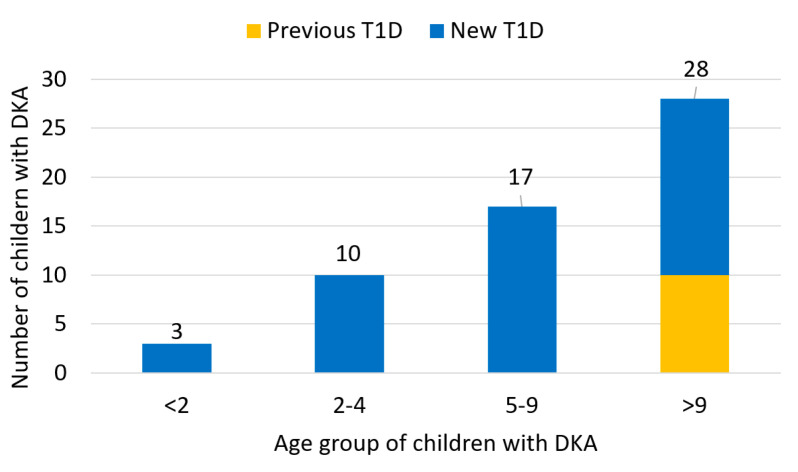
Number of children with diabetic ketoacidosis admitted in Pediatric Intensive Care Unit in the period 2011–2020, divided by age groups.Legend: T1D—type 1 diabetes; DKA—diabetic ketoacidosis.

**Figure 3 medicina-58-00638-f003:**
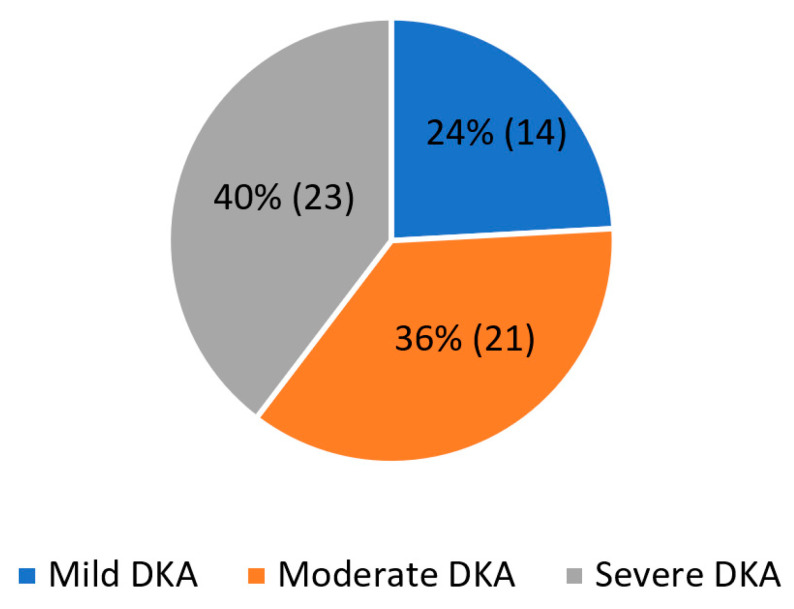
Number and percentage of children with diabetic ketoacidosis admitted in Pediatric Intensive Care Unit in the period 2011–2020, divided by severity of acidosis. Legend: DKA—diabetic ketoacidosis.

**Figure 4 medicina-58-00638-f004:**
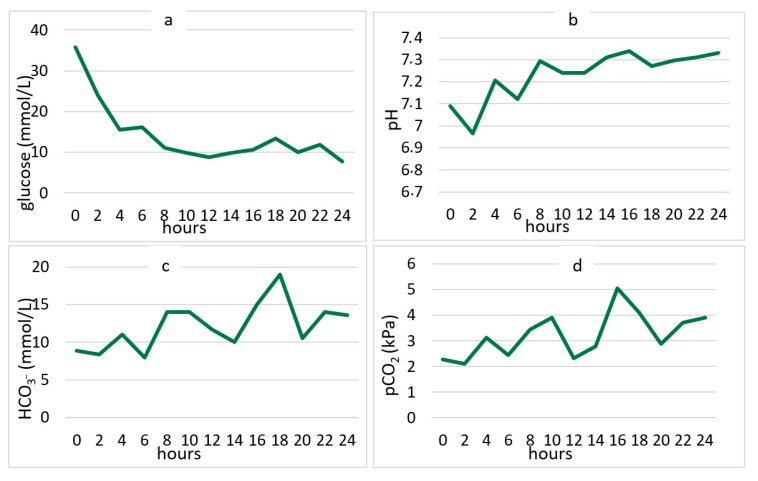
Average changes in blood glucose concentration (**a**), pH values (**b**), bicarbonate concentrations (**c**), and carbon dioxide partial pressure (**d**), for the children with diabetic ketoacidosis during the first 24 h of hospitalization in the Pediatric Intensive Care Unit. Legend: HCO_3_^−^ bicarbonate concentration; pCO_2_—partial pressure of carbon dioxide.

**Table 1 medicina-58-00638-t001:** Demographic, clinical and laboratory characteristics of children with diabetic ketoacidosis admitted to the Pediatric Intensive Care Unit at Clinical Hospital Centre Rijeka, Croatia, in the period 2011–2020.

		Patients with DKA ^a^	
All patients *n* (%)		58 (100%)	
Demographic data	Females	29 (50%)	
	Family history of T1D ^b^	3 (5.8%)	
	Family history of T2D ^c^	25 (48.1%)	
Symptoms at admission	Polydipsia	43 (74.1%)	
	Polyuria	41 (70.7%)	
	Kussmaul breathing	33 (56.9%)	
	Weight loss	29 (50%)	
	Fatigue	29 (50%)	
	Vomiting	25 (43.1%)	
	Acetone breath	24 (41.4%)	
	Loss of appetite	23 (39.7%)	
	Nocturia	23 (39.7%)	
	Abdominal pain	17 (29.3%)	
	Nausea	9 (15.5%)	
Level of consciousness	Alert	45 (77.6%)	
	Respond to Voice	9 (15.5%)	
	Respond to Pain	3 (5.2%)	
	Unconscious	1 (1.7%)	
		SD ^d^	Reference Range
Laboratory parameters	Glucose (mmol/L)	25.05 ± 8.85	3.3–5.5
	pH	7.11 ± 0.18	7.35–7.45
	HCO_3_ ^e^ (mmol/L)	4.7 ± 4.33	21–28
	pCO_2_ ^f^ (kPa)	2.68 ± 3.05	4.5–6.2
	BE ^g^	−20.67 ± 7.11	(−4)–(+2)
	HbA1c ^h^ (%)	12.1 ± 2.21	<6
	Na ^i^ (mmol/L)	132.25 ± 4.28	134–144
	K ^j^ (mmol/L)	4.20 ± 0.67	3.3–4.6
	BUN ^k^ (mmol/L)	5.10 ± 2.93	1.8–6.4
	Creatinine (umol/L)	54 ± 27.7	35–104

Legend: ^a^ diabetic ketoacidosis; ^b^ type 1 diabetes; ^c^ type 2 diabetes; ^d^ standard deviation; ^e^ serum bicarbonate; ^f^ partial carbon dioxide pressure; ^g^ base excess; ^h^ glycated hemoglobin; ^i^ sodium; ^j^ potassium; ^k^ blood urea nitrogen. Data are *n* (%), mean ± standard deviation as appropriate.

## Data Availability

Not applicable.

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
