# Peer review of "Epidemiologic Characteristics of Children with Diabetic Ketoacidosis Treated in a Pediatric Intensive Care Unit in a 10-Year-Period: Single Centre Experience in Croatia"

_medicina, 2022, doi:10.3390/medicina58050638_

Round 1

Reviewer 1 Report

I appreciate to review this manuscript, I have some considerations.

  1. This study does not assess incidence DKA, this is the only descriptive study, there are others statist tests to analyse incidence, please re-write the objective, results and conclusions.
  2. The introduction section is clearly  written.
  3. The discussion section is pertinent to the results.  

Reviewer 2 Report

I would like to thank the authors for this well-presented retrospective study on DKA in their region. Some minor comments:

  1. Line 2: …Unit in a 10-year-period
  2. Line 16: This poses a higher…
  3. Line 21: Data and not The data
  4. Line 28: Polyuria, polydipsia, and Kussmaul breathing are signs rather than symptoms
  5. Line 30: a rising trend
  6. Line 32: of children with diabetes and not “of diabetic children”
  7. Line 49: of varying degrees (and not “to”)
  8. Line 100: first sentence needs to be rephrased. In addition, authors need to erase a full stop that is present after each "2020" written
  9. Line 115: number of children and not “of the children”. Same in line 116
  10. Line 117: In the previous and not “In the remaining”
  11. Line 122: Number of children <2 years presenting with DKA should be expressed as a percentage since, the younger the age of a child with newly diagnosed diabetes, the higher the probability of him/her presenting with DKA
  12. Line 198: need to change to “Two” the number of children that recovered completely
  13. Line 204: “respectively” should be transferred at the end of the sentence
  14. Line 206: need to erase full stop after 2003. Same, in line 212
  15. Line 208: “correlates”
  16. Line 237: A recent meta-analysis showed a significant increase in the incidence of reported DKA (Pala L et al, Acta Diabetol. 2019;56:973). I think this data is more robust and you should rephrase your statement
  17. Line 240: prepubertal, pubertal and/or postpubertal age (not adolescent age)
  18. Line 252: a late clinical sign (not symptom)
  19. Line 264: that in your region if T1D (there are two “that” in the sentence)
  20. Line 268: Since many of your patients were treated before 2018, you should add that ISPAD 2018 guidelines (and corresponding previous versions) were used
  21. Line 298: The first sentence refers to a single limitation and should therefore be appropriately rephrased
  22. Line 309: Most children in western Croatia (“the” should be omitted)

Reviewer 3 Report

Introduction: what would be the research novelty beyond the description of clinical experience?

Methods: how generalisable are the results for county level?

Changes over the time should be statistically tested. Also, the same about Covid-19 impact hypothesis (e.g., joinpoint regression analysis).

Regarding the "Stress is known to be one of possible risk factors for developing TD1"- reference?
